Applied and Environmental Science

# The Stringent Stress Response Controls Proteases and Global Regulators under Optimal Growth Conditions in *Pseudomonas aeruginosa*

Daniel Pletzer,[a,b] Travis M. Blimkie,[a] Heidi Wolfmeier,[a] Yicong Li,[a] Arjun Baghela,[a] Amy H. Y. Lee,[a,c] Reza Falsafi,[a] Robert E. W. Hancock[a]

aCentre for Microbial Diseases and Immunity Research, Department of Microbiology and Immunology, University of British Columbia, Vancouver, Canada
bDepartment of Microbiology and Immunology, University of Otago, Dunedin, New Zealand
cDepartment of Molecular Biology and Biochemistry, Simon Fraser University, Burnaby, Canada

**ABSTRACT** The bacterial stringent stress response, mediated by the signaling molecule guanosine tetraphosphate, ppGpp, has recently gained attention as being important during normal cellular growth and as a potential new therapeutic target, which warrants detailed mechanistic understanding. Here, we used intracellular protein tracking in *Pseudomonas aeruginosa* PAO1, which indicated that RelA was bound to the ribosome, while SpoT localized at the cell poles. Transcriptome sequencing (RNA-Seq) was used to investigate the transcriptome of a ppGpp-deficient strain under nonstressful, nutrient-rich broth conditions where the mutant grew at the same rate as the parent strain. In the exponential growth phase, the lack of ppGpp led to >1,600 transcriptional changes (fold change cutoff of ±1.5), providing further novel insights into the normal physiological role of ppGpp. The stringent response was linked to gene expression of various proteases and secretion systems, including *aprA*, PA0277, *impA*, and *clpP2*. The previously observed reduction in cytotoxicity toward red blood cells in a stringent response mutant appeared to be due to *aprA*. Investigation of an *aprA* mutant in a murine skin infection model showed increased survival rates of mice infected with the *aprA* mutant, consistent with previous observations that stringent response mutants have reduced virulence. In addition, the overexpression of *relA*, but not induction of ppGpp with serine hydroxamate, dysregulated global transcriptional regulators as well as >30% of the regulatory networks controlled by AlgR, OxyR, LasR, and AmrZ. Together, these data expand our knowledge about ppGpp and its regulatory network and role in environmental adaptation. It also confirms its important role throughout the normal growth cycle of bacteria.

**IMPORTANCE** Microorganisms need to adapt rapidly to survive harsh environmental changes. Here, we showed the broad influence of the highly studied bacterial stringent stress response under nonstressful conditions that indicate its general physiological importance and might reflect the readiness of bacteria to respond to and activate acute stress responses. Using RNA-Seq to investigate the transcriptional network of *Pseudomonas aeruginosa* cells revealed that >30% of all genes changed expression in a stringent response mutant under optimal growth conditions. This included genes regulated by global transcriptional regulators and novel downstream effectors. Our results help to understand the importance of this stress regulator in bacterial lifestyle under relatively unstressed conditions. As such, it draws attention to the consequences of targeting this ubiquitous bacterial signaling molecule.

**KEYWORDS** ppGpp, *relA*, *spoT*, *aprA*, global transcriptional regulator, global regulatory networks, proteases, stringent response

Address correspondence to Daniel Pletzer, daniel.pletzer@otago.ac.nz, or Robert E. W. Hancock, bob@hancocklab.com.

Investigation of the Pseudomonas aeruginosa stringent stress response under non-stressful conditions indicate its general physiological importance and might also reflect the readiness of bacteria to respond to and activate acute stress responses.

To deal with stress and/or harmful environmental conditions, microbes can adopt versatile adaptive lifestyles. To enable such lifestyle changes to occur rapidly, bacteria have evolved complex hierarchical regulatory networks to trigger diverse molecular responses that alter gene expression and protein activity. Global regulatory systems enable the coordination of downstream effectors that help recognize and appropriately respond to new environments. In particular, microbial life depends on the ability to rapidly switch from favorable conditions, such as rapid growth in nutrient-rich media, to recognize and counteract external threats and switch into a survival mode (1). Here, we wondered whether such stress adaptations might also operate under optimal, rapid-growth conditions that are not usually considered stressful.

As long as sufficient and appropriate nutrients are provided and toxic agents are absent, bacteria continue to replicate, although in normal culture they eventually stop growing (e.g., in *Escherichia coli* at around two billion bacteria per ml). On a cellular and molecular level, the processes that they undergo during rapid growth are, however, likely quite stressful, with rapid replication, protein synthesis, cell division, and reorganization of the cell (2). For example, there is a disconnect between bacterial division every 20 to 40 min under optimal conditions and replication and segregation of the chromosomal DNA, which is 1,000 times the length of the cell and, thus, highly condensed, that requires 60 to 90 min (2). Furthermore, as the density of bacteria increases, they start to experience depletion of one or more essential nutrients/growth requirements and/or the formation of inhibitory products, such as organic acids, which eventually leads to the stationary phase (3). It is known that maintenance of bacteria in stationary phase is guided by the alternative stress/starvation sigma factor ($\sigma^S$) (4, 5) and the stringent stress response (3), which indicates that the cessation of growth in broth culture occurs under stressful circumstances. However, it is worth asking about the mechanistic impacts of such factors during rapid, apparently uninhibited growth.

One major mechanism for dealing with stress is the stringent stress response intermediated by the second messenger guanosine tetraphosphate (ppGpp) (6, 7). The activation of the stringent stress response during amino acid starvation is due to the accumulation of uncharged, deacylated tRNA molecules in the cytosol that enter the ribosome A site and ultimately cause ribosome stalling (8). In most Gram-negative bacteria, two enzymes, RelA and SpoT, mediate ppGpp homeostasis. Recently, Winther et al. (9) showed that RelA binds to empty tRNA molecules in the cytosol and the tRNA-RelA complex further loads into the A-site of the ribosome. RelA becomes activated and synthesizes ppGpp after interaction with the large Sarcin-Ricin loop of the 23S rRNA (9). On the other hand, SpoT, a bifunctional enzyme with weak ppGpp synthetase activity as well as a ppGpp degradative function, controls the balance of cellular ppGpp levels through its hydrolase activity (10). SpoT is mainly regulated by other environmental stress and starvation signals, including carbon, phosphate, iron, or fatty acid starvation (7). The stringent stress response leads to the dysregulation of a third or more of bacterial genes, enabling stressed cells to divert resources away from growth and division and toward stress coping mechanisms to promote survival until nutrient conditions improve (7).

Mutants lacking RelA and SpoT are unable to produce ppGpp and consequently have multiple defects in coping with stress and show reduced virulence in animal models (11–15). Interestingly, during exponential growth, basal levels of ppGpp in *Bacillus subtilis* act as one of the major regulators of GTP homeostasis (16), while in *E. coli* ppGpp influences negative supercoiling (17) and modestly affects bacterial growth rates in Luria Broth but not M9 glucose minimal medium (18). Recently, ppGpp has been shown to bind to two separate sites on the RNA polymerase, (i) the $\beta'$-$\omega$ subunit and (ii) the interface between the $\beta'$ subunit and the transcription factor DksA. DksA works in concert with ppGpp as a coregulator that amplifies the impact of ppGpp (19). During starvation, cells accumulate ppGpp that interferes with $\sigma^{70}$ binding to promoters and consequently allows for the binding of alternative factors, such as $\sigma^S$ and $\sigma^{32}$, that can redirect the RNA polymerase to energy-saving processes (20, 21). The fact that ppGpp has been shown to interfere with binding of a variety of $\sigma$ factors to the core

RNA polymerase (21, 22) raises the question as to what happens at the transcriptional level when ppGpp is absent. We hypothesized that the absence of a regulatory element impacting RNA polymerase should have substantial transcriptional consequences even under nonstressful conditions. This was further supported by the recent work of Sanchez-Vazquez et al. (23) that demonstrated that overproduction of ppGpp in *E. coli*, after only 5 min and in the absence of any obvious stress, led to the altered expression of more than 750 transcripts dependent on ppGpp binding to the RNA polymerase.

Here, we investigated the promoter activity of the *relA* and *spoT* genes and examined their intracellular localization in *Pseudomonas aeruginosa* PAO1. We used a stringent response mutant and compared its transcriptional landscape to wild-type expression during an unstressed (rich medium) bacterial lifestyle where cells would be likely to mainly focus on growth and replication. Our findings strongly suggest that, in addition to its function in managing stress and growth rate, the stringent response also plays an important role during normal growth.

## RESULTS AND DISCUSSION

**RelA-YFP was observed in a helical arrangement in *P. aeruginosa*.** Bacterial growth rate is regulated and controlled in part by the availability and quantity of ribosomes and rRNA. In *E. coli*, ribosomal gene expression is controlled by ppGpp, which helps to maintain the association of charged tRNAs with the ribosome (24). In *E. coli*, Nakagawa et al. (25) and Sarubbi et al. (26) demonstrated that *relA* gene expression is under dual promoter control, with one promoter constitutively expressed and one activated temporarily during the transition from exponential to stationary growth phase. Here, we confirmed the dual-promoter activity of *relA* in *P. aeruginosa* PAO1 and the requirement of both promoters for maximum expression of RelA during amino acid starvation and in the stationary phase (see Data Set 1 in the supplemental material). At the translational level, the current working model indicates that RelA becomes activated when uncharged tRNA molecules enter the A-site of the ribosome. Activation of RelA and, thus, synthesis of ppGpp, leads to the deactivation and release of RelA from the ribosome until it is reactivated by another stalled ribosome, a mechanism known as hopping between stalled ribosomes (8). This paradigm for the association between the ribosome and RelA has recently been challenged, and it has been suggested that the interaction between RelA and uncharged tRNA does not necessarily involve the ribosome. Nevertheless, activation of RelA only occurs when a RelA-tRNA complex enters the A-site of a stalled ribosome (9). This is supported by cryo-electron microscopy structures of ribosome-bound RelA-tRNA complexes, shown by Li et al. (27) and others (28, 29), where they revealed that RelA synthesizes ppGpp only when bound to the 70S subunit of the ribosome.

This prompted us to further investigate, in *P. aeruginosa*, the subcellular localization of RelA when fused to yellow fluorescence protein (YFP). At single-cell resolution, we identified the helical spatial distribution of RelA-YFP (twisted with alternating bands around the DNA-rich nucleoid area) (Fig. 1A), similar to that observed for ribosomes in *E. coli* (30–32), indicating that, as for *E. coli*, RelA was likely bound to the ribosomes in *P. aeruginosa* when overexpressed. Our data were also in accordance with those of English et al. (33), where they used in single-molecule experiments to show that RelA was bound to the ribosome under nonstarvation conditions. We further investigated the subcellular localization of RelA-YFP upon stress induction using either serine-hydroxamate (SHX) to mimic amino acid starvation or stationary-phase growth of *P. aeruginosa* cultures. While a very similar distribution of RelA-YFP was observed after stress induction, there was a significantly higher coefficient of variation (Fig. 1C), consistent with increased fluorescence distribution as spots within the cell (as opposed to equally distributed signal). This was consistent with an interpretation that RelA was bound to the ribosome but dissociated to some extent during stress.

**SpoT-GFP localized at the cell pole in *P. aeruginosa*.** We utilized a SpoT-GFP fusion protein to visualize its subcellular localization and found that under normal growth conditions, the fusion protein localized at the cell pole and the septal ring of

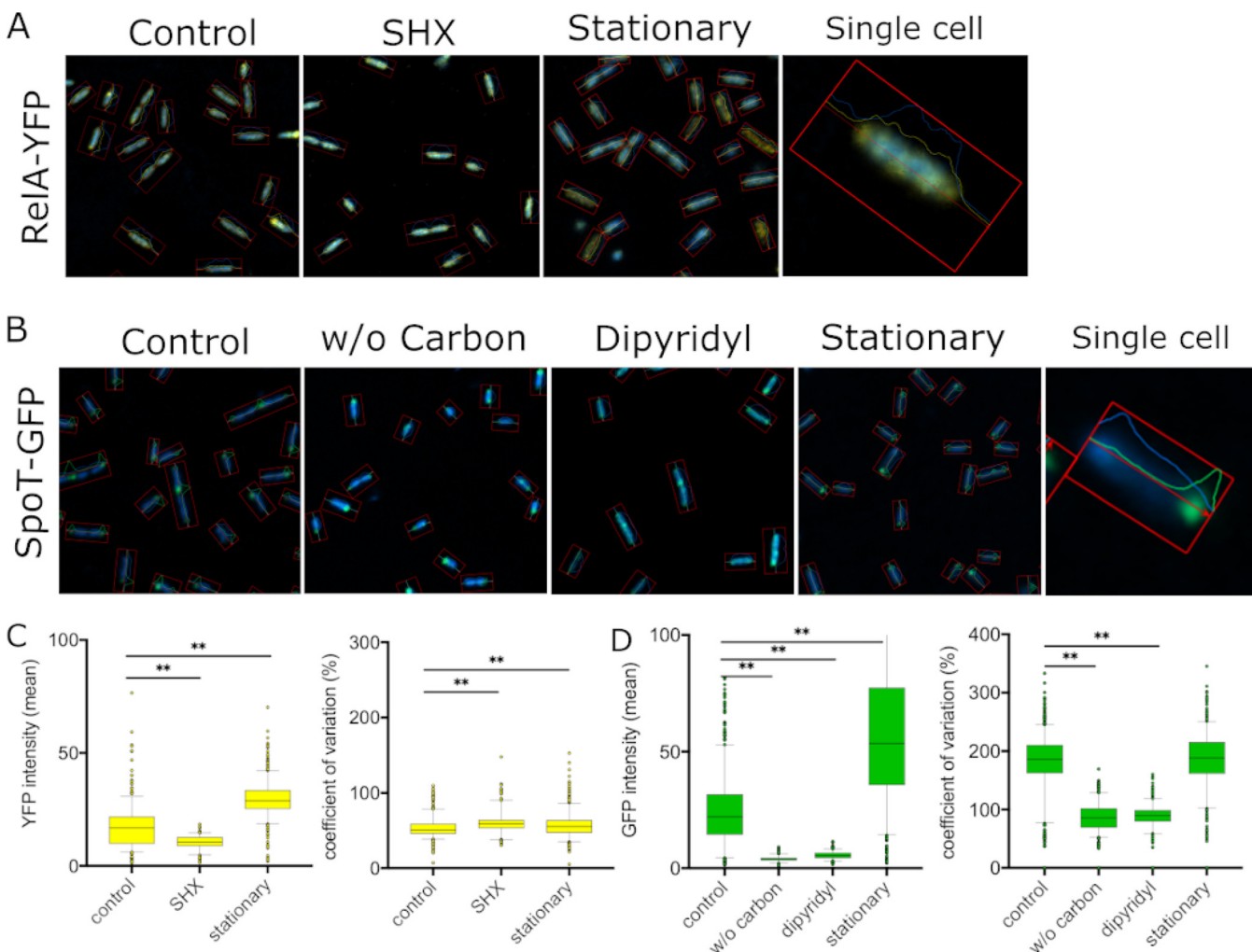

**FIG 1** Subcellular localization of RelA-YFP and SpoT-GFP fusions in *P. aeruginosa* PAO1. Cells were grown in BM2 to mid-exponential phase, and fluorescent constructs were induced with 5% L-arabinose for 30 min prior to individual stress treatments. Cells were washed in PBS, and subsequently the DNA was counterstained with Hoechst 3342 DNA stain (blue). (A) RelA-YFP cultures were treated with or without (control) 1 mM SHX at an OD of 0.5 for 30 min. Stationary-phase cultures were grown to an OD of 2.5 ± 0.3 before induction with 5% L-arabinose. (B) SpoT-GFP cultures were transferred into BM2 (control) or BM2 without carbon source or were treated with 1 mM of the iron chelator dipyridyl at an OD of 0.5 for 30 min. Stationary-phase cultures were grown to an OD of 2.5 ± 0.3 before induction with 5% L-arabinose. (A and B) The pictures to the very right show a single cell under uninduced (control) conditions. The rectangle indicates the identification of single cells that have been used to measure fluorescence intensity. (C) RelA-YFP fluorescence intensity (left) and coefficient of variation (right) with or without (control) 1 mM SHX and in stationary phase. (D) SpoT-GFP fluorescence intensity (left) and coefficient of variation (right) without stress (control) and upon removal of the carbon source, addition of 1 mM iron chelator dipyridyl, and stationary phase. (C and D) Statistics were performed using one-way analysis of variance (ANOVA) with Dunn's multiple-comparison test; $P < 0.01$ (**). The percent coefficient of variation is standard deviation (intensity) divided by mean (intensity) times 100. The higher the coefficient percentage, the higher the likelihood of individual spots in the cells, while a lower percentage indicates a rather equally distributed signal. Rel-YFP ($n = 520$ to 1,700 cells) and SpoT-GFP ($n = 750$ to 1,500 cells) were used.

elongated cells (Fig. 1B). Bacterial cells are well-organized factories where cellular asymmetry and compartmentalization play a vital role in many cellular processes (34). In recent years, the importance of localization of certain proteins at the pole of rod-shaped bacteria has become increasingly evident, with such proteins being crucial for fundamental cellular and regulatory processes (including chromosome segregation, cell cycle, chemotaxis, adhesion, and motility) as well as virulence (34, 35). Intriguingly, the manifestation of pili or flagella at a specific pole allows bacteria to quickly move through the environment and associate with, e.g., mucosal surfaces, which is interesting given the polar localization of SpoT and the role of ppGpp in such processes (36, 37). The localization at the division septum was consistent with results in *Caulobacter*, where SpoT was found to be involved in blocking the initiation of DNA replication and regulation of DnaA, a conserved replication initiator protein (38), although we did not investigate this potential function here.

Intriguingly, upon stress induction by either carbon or iron (dipyridyl treatment) limitation, the median signal intensity of the SpoT-GFP complex decreased by 5.7-fold and 4-fold, respectively, and the signal became more uniformly distributed. The decreased coefficient of variation correlated with irregular fluorescence observed across the cells (Fig. 1B and D). This observation might relate to the ppGpp hydrolyzing activity of SpoT, which presumably maintains ppGpp homeostasis; indeed, inside the cell, ppGpp molecules are broadly distributed and bind to ribosomes that are found around the nucleoid and at the cell poles (30).

**Transcriptional changes dependent on ppGpp occurred during normal growth.** Since both RelA and SpoT were obviously present during logarithmic growth under nutrient-rich conditions (Fig. S1 and S2) (17, 18), we investigated a role for ppGpp under such circumstances. Interestingly, there were no obvious or significant differences, for the studied strains, on growth during the logarithmic phase in three different media (Fig. S3), which contrasted with results in *E. coli* (18), where they used a different growth methodology to avoid spontaneous suppressor mutants. To understand underlying mechanisms, Gaca et al. (39) used microarrays to show that a ppGpp-deficient strain, in the Gram-positive bacterium *Enterococcus faecalis*, altered expression in the exponential phase of growth of 246 genes, including genes influencing pyruvate production and GTP homeostasis. Here, we utilized the more comprehensive method of transcriptome sequencing (RNA-Seq) to test the effect that the absence of ppGpp had on the transcriptome under nutrient-rich, nonstressful conditions in *P. aeruginosa*.

RNA-Seq was performed on the *P. aeruginosa* PAO1 wild type and a strain lacking the ability to produce ppGpp ($\Delta relA$ $\Delta spoT$ double mutant) during exponential growth (optical density at 600 nm [$OD_{600}$] of 0.5) under nutrient-rich conditions (double yeast tryptone [2YT] broth). This revealed 1,673 dysregulated genes with a fold change of ±1.5 and an adjusted (for false discovery) $P$ value of <0.05 (Data Set S1). Expression of a subset of 15 genes (with a fold change around the ±1.5 cutoff) was validated using quantitative real-time-PCR (qRT-PCR), demonstrating similar expression trends and an overall $R^2$ correlation coefficient of 0.66 ($P$ < 0.01) (Fig. S4).

Gene Ontology (GO) and pathway enrichment (KEGG) were used for functional enrichment analysis of differentially expressed (DE) genes identified when comparing the PAO1 stringent response mutant to the PAO1 wild type. This demonstrated the upregulation of six pathways (including biosynthetic and metabolic processes, as well as the type 3 secretion system [T3SS] and cell surface signaling pathways) and downregulation of six pathways (including cell transport, type 2 and 6 secretion systems, and acetyltransferase activity) in GO (Fig. 2A), while KEGG analysis indicated the downregulation of six pathways (including chemotaxis, quorum sensing, and amino acid and fatty acid metabolism) (Fig. 2B). Upregulation, in the stringent response mutant, of the type 3 secretion system was interesting, since this system requires bacterial cells to contact with the host to directly inject its substrates that include cytotoxins (40). In contrast, several genes encoding type 2 secretion systems (PA3095 to PA3103, Xcp genes that mediate exoprotease secretion and PA0677 to PA0689, Hxc genes, and effector LapA) and type 6 secretion systems (T6SS) (Fig. 2A) were downregulated in the mutant. The T6SS, found on three different loci on the chromosome (H1-T6SS, PA0074 to PA0091; H2-T6SS, PA1656 to PA1671; and H3-T6SS, PA2359 to PA2371) target other bacteria by secreting effector proteins that can inhibit or kill them through a contact delivery system (41). The H2 and H3 systems are important *P. aeruginosa* pathogenesis mediators (42). Although GO analysis indicated that the T6SS were overall downregulated in the stringent response mutant (Fig. 2A), the H1 system was actually upregulated. This agrees with others who showed that H1-T6SS is repressed when quorum sensing is highly active, whereas such circumstances activate H2 and H3 (42). Indeed, regulation of the T6SSs might be under quorum-sensing control, since quorum-sensing pathways were downregulated in our KEGG analysis (Fig. 2B).

Quorum sensing is a complex network that cells use for cell-to-cell communication. The quorum-sensing network follows a hierarchy with LasR at the top directly influencing transcriptional regulators, such as *qscR*, *vqsR*, and *rhlR*, which regulate another

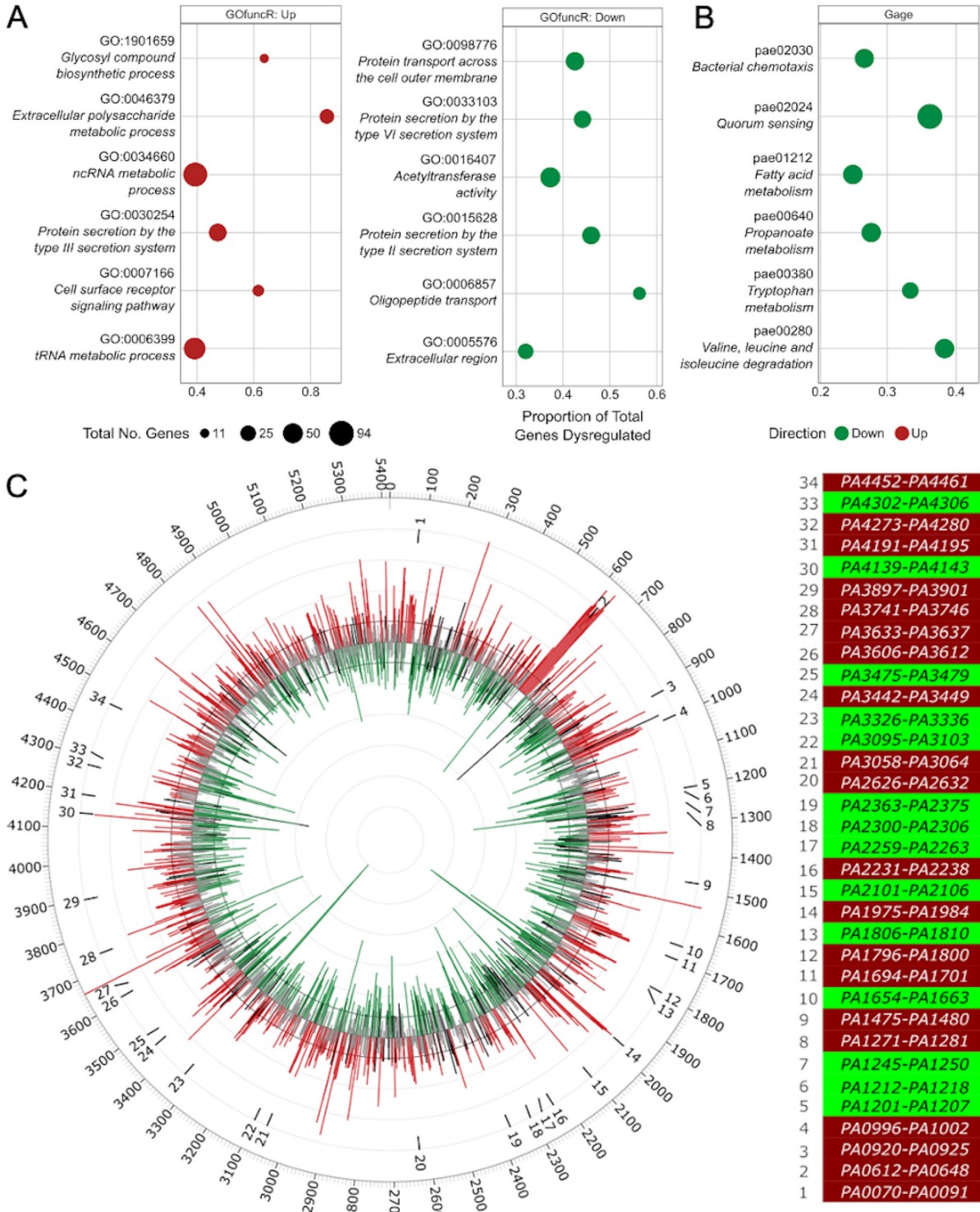

**FIG 2** GO enrichment, KEGG analysis, and circular visualization of the differential expressed genes between the *P. aeruginosa* PAO1 Δ*relA* Δ*spoT* stringent stress response mutant and wild-type strain under planktonic conditions. (A) Results of GO term enrichment performed by GOfuncR on the list of differentially expressed genes, upregulated (left; red) and downregulated (right; green). GO terms were considered significant with a *q* value of ≤0.2. (B) Results of KEGG enrichment performed by GAGE analysis with a threshold of a *q* value of ≤0.2. (A and B) Dot size indicates total number of genes annotated to a particular term/pathway. Gene ratio represents the proportion of total genes assigned to a term that are differentially expressed. (C) Neighboring genes within the DE genes across the chromosome (left). The inner track of the Circos plot shows log$_2$ fold change (gray). All significant values with adjusted *P* value of <0.05 are highlighted in red (log$_2$ fold change, >0.585) and green (log$_2$ fold change, <−0.585). The numbers around the expression pattern show identified regions where a minimum of five consecutive genes were dysregulated in the same direction. The outer track shows the location of the genes (PA0001 to PA5420) on the PAO1 chromosome without the PA prefix. Heatmap of the 34 regions color-coded based on up- (red) or downregulation (green) of the set of the five consecutive dysregulated genes (right). See Data Set S2 for the full list of genes and annotation.

set of regulators, such as two-component response systems (e.g., *gacAS* and *pprAB*), which in turn further regulate quorum sensing (43, 44). The stringent response is known to activate quorum sensing in *Pseudomonas* through increased expression of *lasR* and *rhlR* when *relA* was overexpressed, indicating that quorum sensing can be activated independently of cell density (45), which is in accordance with our data.

We also observed that differential expression often involved coexpressed neighboring genes. This could be visualized by projecting differentially expressed genes onto the circular chromosome (Fig. 2C), which revealed 34 clusters of genes (total of 289 genes, 17% differentially expressed genes) with multiple genes with a similar direction of expression that had a broad array of functions. Indeed, within these clusters, we observed differential expression of genes involved in various secretion systems, transporters, quorum sensing, pyocin synthesis, vitamin B12 synthesis, heme export, cytochrome *c* synthesis, ethanol oxidation, sulfur and carbon metabolism, biofilm formation, fatty acid and lipopolysaccharide biosynthesis, polyamine transport, adherence, ribosomes, and other biosynthetic gene synthesis systems (Data Set S2). Intriguingly, only genes in 8 out the 34 clusters were organized in operons.

**The stringent response is required for environmental adaptations.** Since the $\Delta relA$ $\Delta spoT$ stringent response double mutant grew normally during the logarithmic phase, we postulated that during normal growth the stringent response controls processes that prepare cells for more stressful situations, including environmental adaptations. Therefore, we examined the RNA-Seq data with this in mind.

Once bacteria have invaded host tissues, they must immediately deal with stresses imposed by host responses. One element assisting the initiation of colonization of *P. aeruginosa* and counteracting host responses is direct cytotoxicity toward host cells that processes epithelial surfaces, enables adhesion, and counteracts the action of phagocytic cells (46). Stringent response mutants demonstrate decreased cytotoxicity toward human bronchial epithelial cells as well as minimal hemolytic activity compared to that of the wild type (12). This was consistent with the finding here that many prominent cytolytic proteases were under the control of the stringent response under rapid growth conditions, as judged by their downregulation in the double mutant, including LasA elastase ($-7.3$-fold), LasB elastase ($-7.8$-fold), protease IV (*piv*; $-3.2$-fold), alkaline protease AprA ($-9.6$-fold), and PA3535 ($-2.1$-fold); similarly, the heat-stable hemolysin/rhamnolipid (synthesized by *rhlABC*, $-4.5$-, $-8.3$-, and $-8.7$-fold, respectively) was substantially downregulated (Data Set S1). This is also generally consistent with the role of the stringent response in adjusting to amino acid deprivation, since extracellular proteases could digest proteins in the environment, creating an additional source of needed amino acids. Nevertheless, protease upregulation by ppGpp under normal growth conditions could be considered preparatory to more stressful circumstances.

Conversely, as mentioned above there was upregulation by about 2-fold of the T3SS machinery and regulatory genes in several adjacent operons (PA1699 to PA1725) (Fig. 2) as well as the effector/toxin *exoT*. Both of these observations could be related in part to stringent regulation of quorum-sensing genes under rapid growth conditions, since we observed downregulation in the $\Delta relA$ $\Delta spoT$ stringent response double mutant of *rhlI* ($-3.0$-fold), *rhlR* ($-2.6$-fold), *lasR* ($-1.6$-fold), *rsaL* ($-3.0$-fold), *pqsH* ($-2.0$-fold), and *pqsL* ($-3.6$-fold; cf. *pqsA-E*, which were 3.9- to 4.6-fold upregulated). Exoproteases and hemolysin are upregulated during quorum sensing, which, in turn, is upregulated by ppGpp, whereas T3SS genes are negatively controlled by RhlI (47).

The ability to respond rapidly is necessary to accommodate sudden environmental changes, and the failure to adapt can have negative or even lethal consequences to the organism. Consistent with a role in preparing for environmental adaptation, the stringent response was shown to be required for swarming motility, rhamnolipid production, adherence, and pyoverdine and pyocyanin production in strain PAO1 (Fig. S5), in accordance with previous studies (12, 37, 48). *P. aeruginosa* cells that encounter semiviscous conditions with a poor nitrogen source are driven to swarm

rapidly across surfaces (49). Intriguingly, while the stringent response double mutant was unable to swarm, we found that, in rich-medium liquid broth, where even the wild type fails to swarm, genes required for swarming, such as LasB (50), were strongly dysregulated in the ΔrelA ΔspoT mutant. Indeed, 29% (60 out of 207) of the genes that are known (PAO1 orthologues from Yeung et al. [49]) to influence swarming motility (49) were either upregulated (30 genes) or downregulated (30 genes) in the ΔrelA ΔspoT stringent response mutant. Included in these were regulators required for normal swarming motility, with five (rhlIR, lasR, rsaL, mvaT, and oruR) that were downregulated in the mutant (i.e., positively regulated by the stringent response) and three that were upregulated (gbuR, phoQ, and nosR). Intriguingly, the two-component response regulator gacA actually suppresses swarming, and mutants in this gene demonstrate hyperswarming phenotypes (49); correspondingly, it was 1.8-fold upregulated in the mutant (i.e., suppressed in the WT by ppGpp). We hypothesize that this large proportion of gene expression changes, especially in regulatory genes, prepares for conditions of rapid adaptation to swarming motility.

Swarming and the expression of flagella are also necessary to prepare cells for cell-to-cell or cell-to-surface adherence (49). The Pseudomonas permeability regulator (PprB, PA4296) controls the tad (tight adherence) locus and positively regulates fimbria assembly (51) and flp pilus assembly; therefore, it is a key regulator required for adhesion (52). The tad locus (PA4302 to PA4306, cluster 33), the type IVb pilin flp, and the usher- and chaperone-encoding cupE1 were >2-fold, 7.1-fold, and 2.8-fold downregulated in the stringent response mutant (Fig. 2C and Data Sets S1 and S2). This is in accordance with results for their corresponding regulator PprB, which was 4.7-fold downregulated. Induction of the stringent response led to the expression of pprB and tadG (Table 1), suggesting that the stringent response regulates pilus assembly via PprB. Adaptation and attachment are the first steps before bacterial biofilm formation, which is also regulated by the stringent response (76). Consistent with this, we found that 31% (228 out of 734) of genes that are known to be required for biofilm formation in Pseudomonas (54) were either upregulated (92 genes) or downregulated (136 genes) in the ΔrelA ΔspoT mutant. Intriguingly, 11 biofilm regulators were also identified with 4 upregulated (including efflux pump repressor mexR, T3SS assembly regulator pcrH and repressor ptrB, and anti-sigma factor vreR) and 7 downregulated (including the two-component sensory protein pprA, the chloramphenicol resistance activator cmrA, and mycobactin siderophore uptake regulator femR). Overall, these data support the proposal that the stringent response regulates rapid adaptation to environmental changes.

**The metalloprotease AprA, a novel downstream effector of the stringent stress response, was required for full virulence.** Since the stringent response has been shown to influence P. aeruginosa infection in multiple models (12, 14, 36, 37, 48, 55) and strongly regulated cytolytic proteases under normal growth conditions, we further investigated the importance of one of these, AprA, in a high-density murine skin infection model. The alkaline protease aprA (PA1249) was 4.1-fold upregulated upon SHX induction and 9.6-fold downregulated in the mutant. The deletion of the aprA gene did not affect lesion sizes (Fig. 3B) or bacterial counts (Fig. 3C) in the abscess. However, after 3 days there was a significant enhancement of survival (77%) of the mice infected with the aprA mutant over the mice infected with the wild-type PAO1 (~33% survival of mice) (Fig. 3D). Therefore, AprA acts as a novel downstream effector of the stringent response and is required for full virulence under high-density infections; previous data have implicated this protein in the regulation of virulence and destruction of host defense systems (56).

Based on this result, we further examined protease-coding genes using the combined lists of peptidases/proteases from MEROPS (57) and the Pseudomonas Genome Database (58). It was found that 37.6% (73 of 194 genes; $P < 0.001$) of genes were significantly dysregulated in the stringent response mutant (Fig. 3A). We then focused on genes that were dysregulated in the double mutant, according to RNA-Seq, and showed an inverse correlation, using qRT-PCR, when ppGpp production was induced by SHX or relA overexpressed (Table 1).

**TABLE 1** Fold changes of *P. aeruginosa* PAO1 mRNA expression[a]

| Locus | Gene | Description or product | Fold change relative to WT PAO1 | | |
|---|---|---|---|---|---|
| | | | ΔrelA ΔspoT | WT + SHX | WT + relA++ |
| **Differential effect of loss (ΔrelA ΔspoT) and overproduction of ppGpp (+SHX/relA++)** | | | | | |
| PA0277 | | Zn-dependent protease with chaperone function | **9.9** | **−5.2** | −1.1 |
| PA0572 | impA | Hypothetical protein | **−9.1** | **2.8** | 1.3 |
| PA1245 | aprX | Type I secretion system | **−2.8** | **4.3** | **2.3** |
| PA1249 | aprA | Alkaline metalloprotease | **−9.6** | **4.1** | **2.0** |
| PA1384 | galE | UDP-glucose 4-epimerase | −1.9 | **4.4** | 1.2 |
| PA1430 | lasR | Transcriptional regulator | −1.6 | −1.1 | **3.0** |
| PA1477 | ccmC | Heme exporter protein | 1.6 | **−2.3** | 1.2 |
| PA1656 | hsiA2 | Type VI secretion protein | −1.8 | 1.2 | 1.8 |
| PA1874 | bapA | Adhesion protein | **−4.3** | **3.0** | **2.0** |
| PA2016 | liuR | Regulator of *liu* genes | −1.6 | **2.0** | −1.4 |
| PA2050 | | RNA polymerase ECF-subfamily σ70 factor | **2.5** | −1.5 | −1.1 |
| PA2227 | vqsM | Transcriptional regulator | −1.7 | 1.4 | **2.5** |
| PA2259 | ptxS | Transcriptional regulator | −1.9 | **2.4** | −1.2 |
| PA2302 | ambE | Nonribosomal peptide synthetase | **−4.1** | **3.0** | 1.4 |
| PA2911 | | TonB-dependent receptor | **8.0** | **−3.0** | 1.3 |
| PA3095 | xcpZ | General secretion pathway protein | **−2.4** | 1.9 | 1.1 |
| PA3326 | clpP2 | ATP-dependent Clp protease proteolytic subunit | **−10.9** | **2.4** | 1.8 |
| PA3327 | | Nonribosomal peptide synthetase | **−21.0** | **8.0** | **2.0** |
| PA3462 | | Sensor/response regulator | −1.5 | **2.2** | 1.1 |
| PA3607 | potA | Polyamine transport protein | **2.9** | **−9.9** | −1.1 |
| PA4210 | phzA1 | Phenazine biosynthesis protein | **−9.8** | **4.2** | **2.5** |
| PA4211 | phzB1 | Phenazine biosynthesis protein | **−7.9** | **12.7** | 1.9 |
| PA4296 | pprB | Two-component response regulator | **−4.7** | **3.8** | 1.7 |
| PA4297 | tadG | Pilus assembly | **−2.0** | **3.1** | 1.4 |
| | | | | | |
| **Smaller effect of either loss of ppGpp or overproduction (+SHX and/or relA++)** | | | | | |
| PA0149 | | RNA polymerase ECF-subfamily σ70 factor | 1.8 | 1.2 | 1.0 |
| PA0625 | | Pyocin production | **6.8** | 1.2 | 1.2 |
| PA0652 | vfr | Transcriptional regulator | −1.7 | 1.1 | 1.2 |
| PA0762 | algU | RNA polymerase σ factor AlgU | 1.9 | −1.2 | 1.0 |
| PA0893 | argR | Transcriptional regulator | 1.1 | −1.0 | **2.1** |
| PA1271 | | TonB-dependent receptor | **3.4** | −1.0 | 1.3 |
| PA1272 | cobO | Cob(I)alamin adenosyltransferase | **2.4** | −1.2 | 1.2 |
| PA1541 | | Drug efflux transporter | **10.9** | **2.4** | 1.0 |
| PA1698 | popN | Type III secretion protein | **3.3** | −1.3 | 1.4 |
| PA2069 | | Carbamoyl transferase | **−8.1** | 1.4 | 1.4 |
| PA2193 | hcnA | Hydrogen cyanide synthase | **−3.2** | −1.5 | −1.1 |
| PA2231 | pslA | Exopolysaccharide biosynthesis | **2.3** | 1.2 | 1.6 |
| PA2360 | hsiA3 | Type VI secretion protein | −1.9 | 1.0 | 1.2 |
| PA2426 | pvdS | RNA polymerase ECF-subfamily σ70 factor | −1.1 | **−18.1** | **−2.7** |
| PA3385 | amrZ | Alginate and motility regulator | −1.0 | 1.0 | **2.1** |
| PA3410 | hasI | RNA polymerase ECF-subfamily σ70 factor | **3.0** | −1.4 | 1.0 |
| PA3661 | | Hypothetical protein | **17.2** | **5.1** | **5.4** |
| PA3899 | fecI | RNA polymerase σ factor | 1.9 | −1.1 | −1.1 |
| PA5261 | algR | Alginate biosynthesis regulatory protein | −1.4 | −1.2 | **2.1** |
| PA5344 | oxyR | Transcriptional regulator | −1.1 | 1.2 | 1.9 |

[a]Shown are the fold changes of *P. aeruginosa* PAO1 mRNA expression in the ΔrelA ΔspoT double mutant compared to the wild type (RNA-Seq data) and compared to the influence of overexpression in the WT of *relA* and induction with 1 mM serine hydroxamate (SHX) (qRT-PCR). Changes greater than 2-fold are presented in boldface.

The zinc-dependent protease PA0277 (5.2-fold downregulated by SHX induction, 9.9-fold upregulated in the mutant) is directly controlled by the posttranscriptional regulator RsmA (59), and the expression of *rsmA* is directly activated by AlgR (60) and indirectly via GacA through RsmY and RsmZ (61). The response regulators AlgR and GacA were controlled by the stringent stress response under normal conditions. Intriguingly, Bowden et al. (62) described a link between the stringent stress response and the loss of *rsmA* expression, which restored protease production in the plant

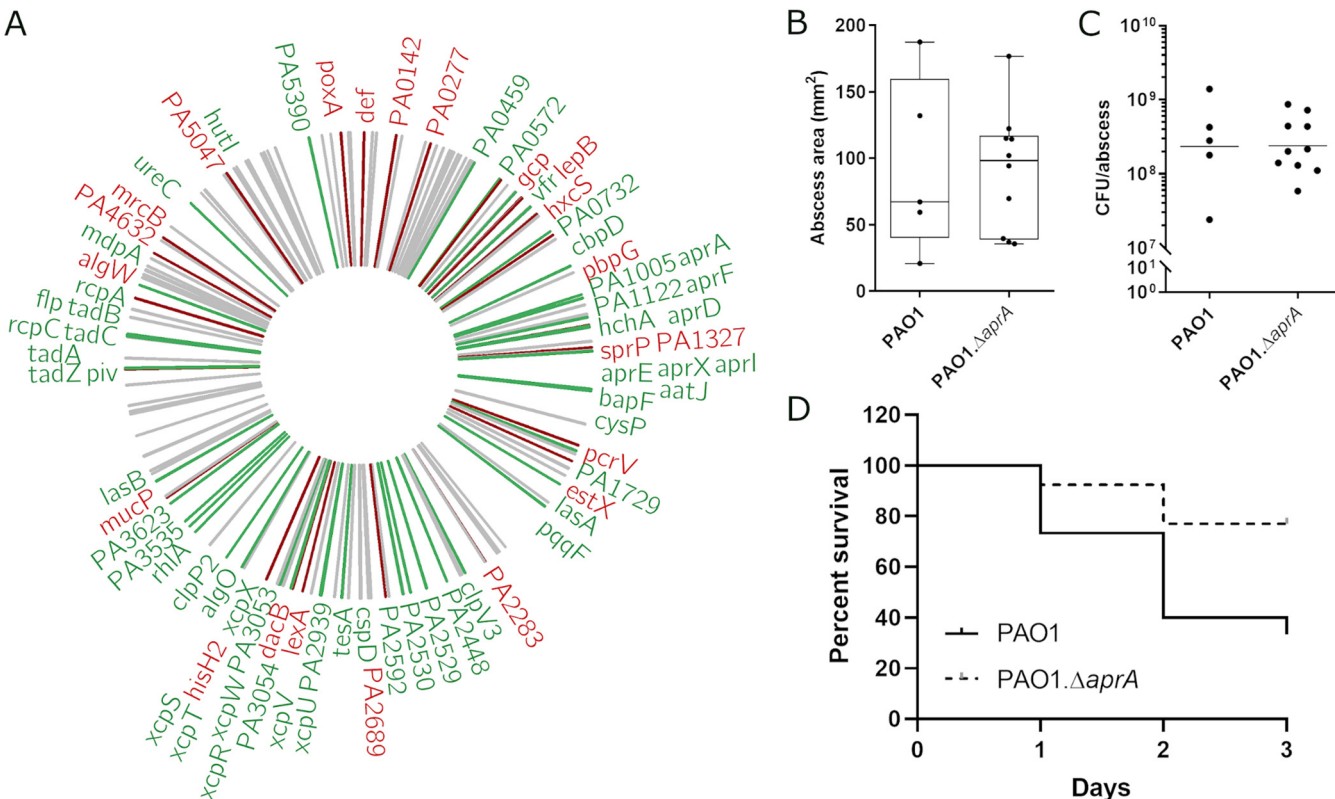

**FIG 3** Differentially expressed protease genes of the stringent response mutant compared to the wild type and alkaline protease mutant *aprA* in a high-bacterial-density skin infection model. (A) Circos plot of proteases and peptidases (www.pseudomonas.com and MEROPS) with downregulated (green) and upregulated (red) genes around the chromosome. Proteases that were not significant are delineated as gray. (B to D) PAO1 wild type (*n* = 15) and PAO1 *aprA*-deficient strain (*n* = 13) in a cutaneous mouse infection model. (B) Abscess sizes. (C) Bacterial load recovered from abscess tissue 3 days postinfection. (D) Survival percentage over the course of a 3-day experiment. Statistics were performed using the Gehan-Breslow-Wilcoxon test (*P* = 0.032) and log-rank (Mantel-Cox) test (*P* = 0.025); a *P* value of ≤0.05 was considered significant.

pathogen *Erwinia atrosepticum*. This further supports our conclusions that proteases are downstream effectors of the stringent response in *P. aeruginosa*.

The immunomodulating metalloprotease *impA* (PA0572) was 2.8-fold upregulated by SHX and 9.1-fold downregulated in the mutant. ImpA is important during infection and protects *P. aeruginosa* from neutrophil attack by cleaving the P-selectin glycoprotein ligand-1 on neutrophils as well as targeting CD43 and CD44, involved in leukocyte homing (63). ImpA contains a LasR-regulated Xcp-dependent signal sequence (64), but SHX induction did not influence the expression of *lasR*. Therefore, *impA* might be under direct control of the stringent response. The putative caseinolytic peptidase *clpP2* (PA3326) was 2.4-fold upregulated by SHX and 10.9-fold downregulated in the stringent response mutant; it is known to be involved in motility, biofilm formation, pigmentation, and iron scavenging (65). Thus, we investigated the impact of these proteases in the high-density murine skin infection model. There was no effect when transposon mutants in PA0277, ImpA, or ClpP2 were tested (data not shown).

**Stringent regulation of global transcriptional regulators.** The large number of differentially expressed genes was apparently controlled in a hierarchical process. Thus, no fewer than 132 regulators were differentially expressed in the Δ*relA* Δ*spoT* double mutant, with 79 being downregulated and 53 upregulated. More highly dysregulated and prominent regulators included alternative sigma factor and regulator of the general stress response and quorum-sensing *rpoS* (−3.8-fold), quorum-sensing regulators *rhlR* (−3.0-fold and −2.6-fold) and *rsaL* (−3.0-fold), the global regulator of virulence and quorum sensing, *vqsR* (−2.5-fold), virulence sigma factor *vreI* (3.0-fold), cold shock protein *cspD* (−3.0-fold), upregulated T3SS repressor *ptrB* (8.1-fold), regulator *pcrG* (2.3-fold), DNA damage-inducible regulator *lexA* (2.3-fold), the alternative

mSystems®

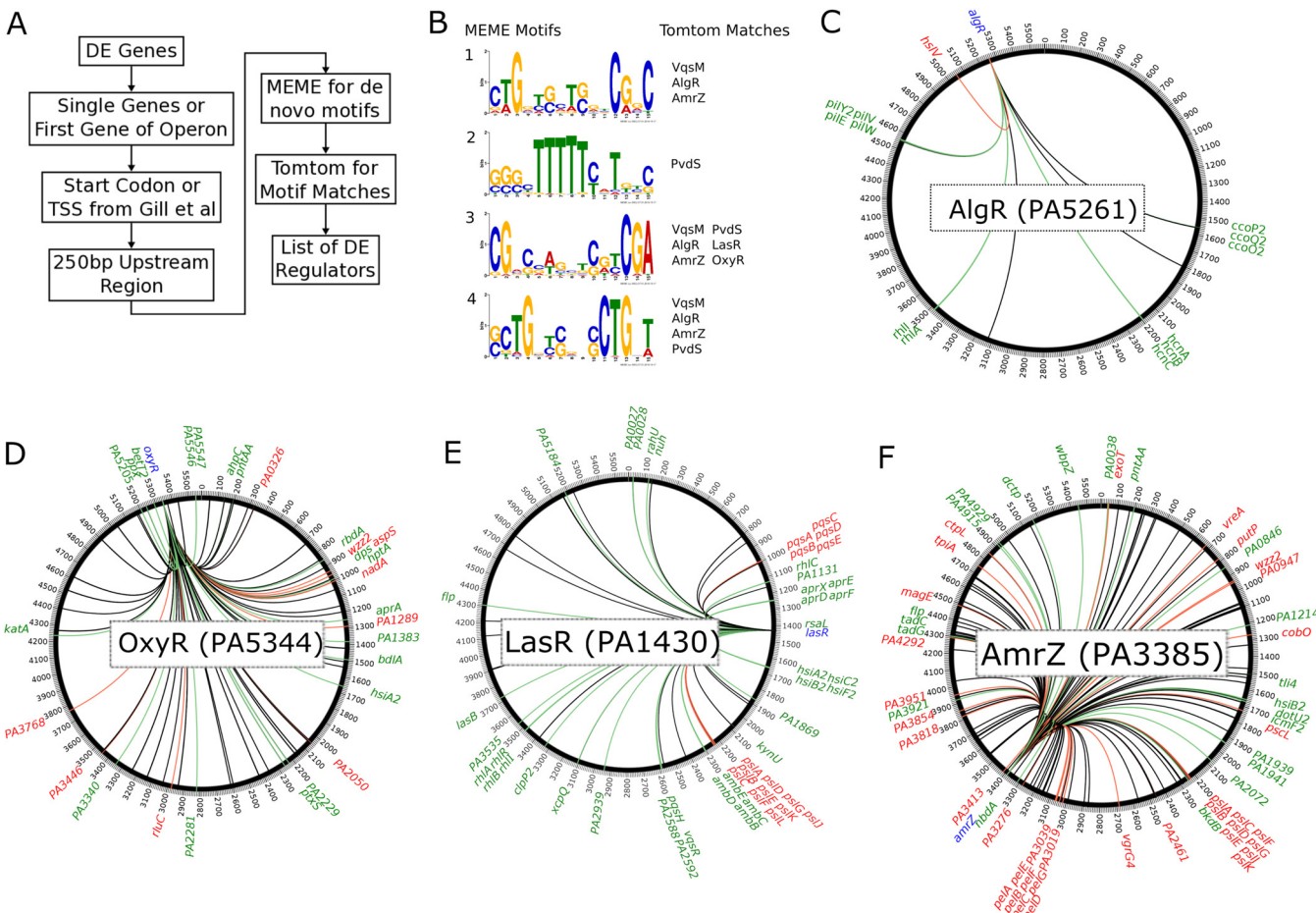

**FIG 4** Motif search and regulons of identified transcription factor binding sites from CollecTF. (A) Overview of the identification pipeline to find dysregulated regulators in RNA-Seq data (66). TSS; transcriptional start site. (B) Identified motifs of dysregulated genes by MEME and regulators by TomTom. (C) AlgR regulon (20 genes; 12 downregulated, 1 upregulated). (D) OxyR regulon (89 genes; 20 downregulated, 10 upregulated). (E) LasR regulon (83 genes; 36 downregulated, 15 upregulated). (F) AmrZ regulon (144 genes; 22 downregulated, 37 upregulated). (C to F) Red, upregulated; green, downregulated; blue, regulator; black, no change. Adjusted *P* value of ≤0.05, fold change of ±1.5. The outer track shows the location of the genes (PA0001 to PA5420) on the PAO1 chromosome without the PA prefix.

sigma factor involved in alginate biosynthesis, *algU* (1.9-fold), the aerobic ethanol oxidation system regulator *erbR* (6.8-fold), two-component system *eraSR* (6.2-fold and 4.1-fold), the nitrous oxide reduction regulator *nosR* (5.4-fold), the efflux pump activator *brlR* (2.4-fold), downregulated two-component regulators of CupE fimbrial assembly activation, *pprBA* (−1.8-fold and −4.7-fold), and PA4074 (−3.2-fold).

The potential that regulators such as these were mediating the very substantial effects of RelA and SpoT during logarithmic growth phase prompted us to further investigate global regulation in the ppGpp-deficient strain. The 1,673 DE genes were used to search for transcriptional regulators (Fig. 4A) that influenced their expression by filtering to retain the first individual differentially expressed in an operon; this reduced the list to 1,201 unique genes. Based on the location of the start of the gene, we then extracted the region 250 bp upstream of the transcriptional initiation sites for these genes (66) and searched for potential transcription factor binding site motifs. Motif-based sequence analysis (MEME) identified four significant motifs (Fig. 4B) that were further processed using the motif comparison tool Tomtom, which revealed several known global regulators from CollecTF, namely, VqsM, AlgR, AmrZ, PvdS, LasR, and OxyR (Fig. 4B). In the next step, the regulon of each regulator extracted from CollecTF (67) was further tested for statistical enrichment in the original list of differentially expressed genes, which revealed four significantly dysregulated regulons, AlgR

($P < 0.02$), OxyR ($P < 0.02$), LasR ($P < 0.01$), and AmrZ ($P < 0.01$); of these, only LasR was modestly 1.57-fold downregulated. The relevance of this analysis is indicated by the fact that global regulators like these control many downstream targets affected by the stringent response, as typified by LasR, a mediator of the 3-oxo-C12-acylhomoserinelactone-mediated quorum-sensing response (45).

**Global transcriptional regulators responded to *relA* overexpression but not SHX induction.** To further explore regulation by the stringent-response mediator, we overexpressed the *relA* gene in the WT and induced the production of ppGpp with SHX (Table 1). Intriguingly, the expression of several transcriptional regulators, including *argR*, *lasR*, *vqsM*, *amrZ*, *algR*, and, to some extent, *oxyR*, was only increased when *relA* was overexpressed but not when the stringent response was induced with SHX (Table 1). SHX treatment triggers stress, since it is basically equivalent to amino acid starvation, while *relA* overexpression should in principle only affect ppGpp levels. These situations are not precisely equivalent, and we would argue that *relA* overexpression more precisely mimics effects during normal growth while SHX reflects an alarm response. Alterations in ppGpp levels due to *relA* overexpression could lead to the shrinkage and depletion of the GTP and c-di-GMP pools that might interfere with transcriptional and translational downstream processes (68), although the added impact of SHX-induced starvation would be a major consideration.

In *P. aeruginosa*, low levels of c-di-GMP can trigger the activation of the global regulators such as AmrZ or AlgR. AmrZ, involved in biofilm production, iron homeostasis, and motility, has been shown to regulate diguanylate cyclases and phosphodiesterases to modulate c-di-GMP (69, 70). Under conditions where ppGpp was not synthesized in the double mutant, we found that ~44% (59 out of 144) of genes of the AmrZ regulon were dysregulated (Fig. 4F), including upregulated genes in polysaccharide production, *psl* (cluster 16) and *pel* (cluster 21), and downregulated genes in the type VI secretion *hsi* (cluster 10) (Fig. 2). A comparison to the chromatin immunoprecipitation sequencing (ChIP-Seq) analysis by Jones et al. (69) further revealed that ~29% of the regulon (115 out of 398) was dysregulated in the stringent response mutant (with 74 genes in common between the two regulons). However, AmrZ-mediated regulation is itself quite complex, since AmrZ belongs to the regulon of the quorum-sensing regulator LasR (71).

In the stringent response mutant, ~61% (51 of 83) of genes in the LasR network were dysregulated (Fig. 4E), including rhamnolipid *rhl* (cluster 25), biofilm matrix polysaccharide *psl* (cluster 16), type VI secretion *hsi* (cluster 10), quinolone quorum-sensing *pqs* (cluster 4), anti-metabolite *amb* (cluster 18), alkaline protease *apr* (cluster 7) (Fig. 2C and 4E), and various other exoenzymes mentioned above. Furthermore, we observed altered expression in the Δ*relA* Δ*spoT* double mutant of 7 of the 10 genes with direct LasR binding sites in their promoters (71), namely, *aprX-A*, *pqsH*, *rhlR*, *rsaL*, *pslA-M*, *ambB-E*, and *clpP2* (Data Set S1). A comparison to the ChIP-Seq analysis by Gilbert et al. (71) showed similar dysregulation of ~64% (47 out of 74) of genes in the LasR regulon (with an overlap of 72 genes in both regulons).

The stringent response has been suggested to affect OxyR and might modify the cellular redox state (72). OxyR uses a redox-sensing mechanism to sense oxidative damage such as that caused by $H_2O_2$ in the cytoplasm in *E. coli* (73). When oxidized, it induces the expression of a catalase that further reduces $H_2O_2$. The stringent response is required for optimal catalase activity (72), which is in accordance with our finding that *katA* (PA4236) was 2.4-fold downregulated in the stringent response mutant. The OxyR regulon also controls quorum sensing, protein synthesis, oxidative phosphorylation, and iron homeostasis (53), and ~34% (30 of 89) of regulon genes were dysregulated in the stringent response mutant (Fig. 4D). Similarly, the double mutant demonstrated altered expression of ~34% (19 out of 56) of genes of the OxyR regulon identified by ChIP-Seq by Wei et al. (53), including *ahpC*, *katA*, *dpS*, PA2050, *pntAA*, *aspS*, and PA1541 (with a total overlap of 44 genes in the two published versions of the OxyR regulon).

AlgR is important for type IV pilus and alginate production and synthesis of c-di-GMP via the diguanylate cyclase MucR (74). Of the small known AlgR regulon in

CollectTF, 65% (13 of 20) of genes were dysregulated in the double mutant (Fig. 4C), including genes encoding pilus, hydrogen cyanide production (*hcn*), and cytochrome *c* oxidase (*cco*). In addition, a ChIP-Seq analysis by Kong et al. (74) revealed a much larger regulon, and we found ~31% (48 of 155) of those genes were dysregulated in the stringent response mutant, only one of which overlapped the CollectTF regulon.

The stress $\sigma$ factor RpoS is interrelated with the stringent response and is known to operate during normal growth (75). Intriguingly, the regulatory networks of RpoS, as well as RpoN, RpoD, and AlgU, were significantly dysregulated in the stringent stress response mutant (Fig. S6), and alternative $\sigma$ factors PA0149, PA0762 (*algU*), PA2050, PA2896 (*sbrl*), PA3410 (*hasl*), and PA3899 (*fecI*) were all upregulated in this mutant (Table 1 and Data Set S1). However, overexpression of *relA* or induction of the stringent response did not influence the expression of any of these $\sigma$ factors, suggesting that their dysregulation in the mutant is because of binding and stability difficulties of the RNA polymerase due to missing ppGpp.

**Conclusions.** In conclusion, we find that the *P. aeruginosa* stringent stress response influences not only global transcriptional regulators under normal growth conditions but also the expression of downstream effectors that potentially enable rapid adaptations. Our findings expand our knowledge about the stringent response and suggest it has an important role throughout the bacterial life. The stringent response, originally discovered as a stress response during amino acid starvation, influences the expression of hundreds of genes (15). It is now clear that the stringent response is more than just a response to starvation and that the production of the signaling molecule ppGpp strongly influences global gene expression of bacterial cells. Here, we further defined the stringent response as being required under nutrient-rich, rapid-growth conditions. This reinforces proposals that the stringent response is an excellent, novel target for the development of new antimicrobials (12, 76, 77). Further mechanistic understanding of downstream processes after stringent response signal blockage will help in the development of novel compounds for clinical use and reveal new targets among the downstream effectors, such as the alkaline protease AprA.

## MATERIALS AND METHODS

**Bacterial strains and growth conditions.** Bacterial strains and plasmids are listed in Table S1A and 1B in the supplemental material. All organisms were cultured at 37°C in double yeast tryptone (2YT) or basal medium 2 (BM2) (78). Liquid cultures were placed with shaking at 250 rpm. Cultures harboring individual vectors were supplemented with 15 $\mu$g/ml gentamicin (Gm) and 100 $\mu$g/ml ampicillin (Ap) for *E. coli* and 50 $\mu$g/ml Gm and 250 $\mu$g/ml carbenicillin (Cb) for *P. aeruginosa*. Bacterial growth was monitored at the optical density at 600 nm ($OD_{600}$) using either a spectrophotometer or a 96-well microtiter plate reader (Synergy H1; BioTek).

**Molecular methods.** PCRs were carried out using Phusion DNA polymerase (Thermo Scientific), in accordance with the manufacturer's instructions, and optimized annealing temperatures for each primer set. PCRs were supplemented with 5% dimethyl sulfoxide. Restriction digestions were performed using Thermo Scientific FastDigest restriction enzymes according to the manufacturer's instructions. All ligation reactions were carried out at room temperature using Thermo Scientific T4 DNA ligase. DNA purifications were performed using the GeneJET PCR purification kit (Thermo Scientific) or the GeneJET gel extraction kit (Thermo Scientific) by following the manufacturer's instructions.

**Construction of the PAO1 arginine *argB* auxotroph and alkaline protease *aprA* mutant.** The construction of both knockout vectors was based on the protocol by Zumaquero et al. (79) and carried out as previously described (80). Briefly, primers argB_up_fwd(Bam)/argB_up_rev and argB_down_fwd/ argB_down_rev(Hind) were used to amplify the 500-bp knockout alleles for the *argB* gene from *P. aeruginosa* PAO1 genomic DNA. Primers aprA_up_fwd(Bam)/aprA_up_rev and aprA_down_fwd/aprA_d-own_rev(Hind) were used to amplify the 500-bp knockout alleles for the *aprA* gene from *P. aeruginosa* PAO1 genomic DNA (Table S1C). The obtained fragments were used in an overlapping PCR with up_fwd and down_rev primers. Next, each fusion fragment was cloned into the suicide vector pEX18Gm via BamHI/HindIII restriction sites and verified by sequencing. The generation of both mutants was based on the site-specific insertional mutagenesis strategy of Schweizer et al. (81) and carried out as described previously (82). To confirm the deletions, locus-specific primers that bind up- and downstream of the amplified knockout alleles were used (argB_outA/argB_outB or aprA_outA/aprA_outB) and the resulting knockout fragments verified by sequencing.

**Cloning of the PAO1 *relA*, *cysM*, and *rpoZ* transcriptional promoter reporter fusions.** Transcriptional fusions between the *relA* promoter region and the *mCherry* gene were created using the primers relA-Pro_fwd(Xba)/relA-Pro_rev-mCherry_fwd to amplify the 84-bp *relA* promoter from PAO1 genomic DNA and mCherry_fwd-relA-Pro_rev/mCherry_rev_t0(Kpn-Apa) to amplify mCherry from plasmid

pUCP23.mCh. The resulting fragments were gel purified and fused in another PCR with relA-Pro_fwd/mCherry_rev primers. The fusion product was cloned onto pUCP22 in the direction opposite that of the lac promoter via XbaI/KpnI restriction sites. Transcriptional fusions between the promoter regions of *cysM* (270 bp) and the *mCherry* gene were created on plasmid pUCP22. Therefore, the mCherry gene was flipped onto pUCP22 via EcoRI/BamHI, yielding mCherry in the direction opposite that of $P_{lac}$. Next, the *cysM* promoter was amplified from PAO1 genomic DNA via cysM-Pro_fwd(Xba)/cysM-Pro_rev(Bam), gel purified, and cloned onto pUCP22.mCherry via XbaI/BamHI (yielding pUCP22.cysM-Pro.mCherry). Transcriptional fusions of the *cysM* promoter, the *relA* promoter, and the *mCherry* gene were created via amplification of the *cysM* promoter (270 bp) from genomic DNA using cysM-Pro_fwd(Xba)/cysM-Pro_rev(Spe). The obtained fragment was ligated via XbaI/SpeI in front of the *relA* promoter on plasmid pUCP22.relA-Pro.mCherry to yield pUCP22.cysM-relA-Pro.mCherry. All created constructs were confirmed by sequencing and transformed into the PAO1 wild type as well as the Δ*argB* auxotroph arginine mutant, as described earlier (82).

Transcriptional fusions between the promoter regions of *rpoZ* and the *eGFP* gene were created on plasmid pUCP23. The *rpoZ* promoter region (88 bp) was amplified via rpoZ-Pro_fwd(SacI)/rpoZ-Pro_rev-egfp_fwd from PAO1 genomic DNA and the *eGFP* gene from pBBR.TIR.egfp.t0 via egfp_fwd-rpoZ-Pro_rev/egfp_rev(Kpn). Both fragments were fused with a subsequent PCR using rpoZ-Pro_fwd/egfp_rev and further transferred onto pUCP23 via SacI/KpnI in the direction opposite that of the lac promoter. All created constructs were confirmed by sequencing and transformed into PAO1 as described earlier (82).

**Promoter fusion constructs: fluorescence and growth curve experiments.** Overnight-grown bacteria (16 to 18 h) in 2YT medium at 37°C with shaking (250 rpm) were washed (8,000 rpm, 3 min), resuspended in BM2, and further diluted to an $OD_{600}$ of 0.1. Two hundred microliters was transferred to a flat-bottom 96-well polystyrene microtiter plate (Corning). Plates were incubated at 37°C with continuous fast linear shaking at 567 cycles per minute (cpm) in a microplate reader (Synergy H1; BioTek). $OD_{600}$ and fluorescence (*mCherry* and *eGFP* were detected at 580 to 610 nm and 488 to 509 nm, respectively) readings were taken every hour over a 16-h period. Experiments were performed three times with at least three technical replicates. Amino acid starvation growth experiments were performed with the PAO1 wild type and the Δ*argB* auxotroph arginine mutant, both carrying promoter fusion constructs, in minimal medium BM2 without $NH_4Cl$ and supplementation of 6.25 μM or 12.5 μM arginine. Statistical analysis was performed using a paired one-sided *t* test where each time point was compared to the starting time point to identify a significant increase in promoter activity.

**Cloning of the inducible *relA* and *spoT* constructs.** The *relA* gene was PCR amplified from PAO1 genomic DNA with primers relA_fwd(Xba)/relA_rev(Hind). The amplified product was gel purified and cloned into pHERD20T via XbaI/HindIII to allow expression from the pBAD promoter and subsequently sequenced. The *spoT* gene was PCR amplified from PAO1 genomic DNA with primers spoT_fwd/spoT_rev(Hind). The amplified product was gel purified, digested with HindIII, and cloned into pHERD20T via SmaI/HindIII to allow expression from pBAD and was subsequently sequenced. Each plasmid was transformed into the *P. aeruginosa* PAO1 wild type as previously described (82).

**Cloning of the RelA-YFP and SpoT-GFP translational reporter fusions.** The *relA* gene was PCR amplified from PAO1 genomic DNA with primers relA_fwd/relA_rev. Both primers had an EcoRI restriction site incorporated. The reverse primer lacked the RelA stop codon. The amplified product was cloned onto pBAD24.yfp via EcoRI and orientation verified via restriction digest and sequencing. The resulting pBAD24.relA-linker-yfp plasmid was digested with BamHI, treated with S1 nuclease, and further digested with HindIII. The resulting fragment was transferred onto pHERD20T via HindIII/SmaI.

The *spoT* gene was PCR amplified from PAO1 genomic DNA with primers spoT_fwd/spoT_rev-link-gfp_fwd. The reverse primer lacked the SpoT stop codon and had an in-frame linker sequence (ATGGTGTCTATCACTAAAGATCAAATC) fused to the forward sequence of *egfp*. The *egfp* gene was amplified from pBBR1.TIR.egfp.t0 with primers gfp-fwd-link-spoT-rev/egfp-rev, whereby the gfp-fwd primer was the reverse complement primer of the spoT-rev primer to allow the fusion of both fragments in a second PCR with spoT_fwd and gfp_rev primers. The amplified product was gel purified, digested with HindIII, and cloned into pHERD20T via SmaI/HindIII before sequencing. Reporter fusions were verified as described below.

**Functional verification of the RelA-YFP and SpoT-GFP reporter fusions using swarming.** Since the fluorescence fusions were in-frame at the C-terminal end of the gene of interest, a fluorescent signal would indicate a correctly folded fusion protein. However, overexpression of fusion proteins can lead to misfolding and subsequent sequestering into insoluble inclusion bodies (83). To verify that both fusion proteins were correctly folded and functional, we complemented the Δ*relA* Δ*spoT* double mutant with either RelA-yellow fluorescent protein (YFP) or SpoT-green fluorescent protein (GFP) and tested the ability of the strain to swarm on semisolid agar plates (BM2 plates containing 0.4% agar). The PAO1 stringent response double mutant is unable to swarm (Fig. S7) (37, 84). The PAO1 wild-type strain as well as the Δ*relA* Δ*spoT* mutant were transformed with an empty pHERD20T vector control and RelA-YFP and SpoT-GFP fusions. All strains were scraped from overnight-grown plates and suspended in sterile demineralized water to an $OD_{600}$ of 0.025. Ten microliters of a bacterial cell suspension was applied onto a swarming agar plate and incubated at 37°C for 18 h. Experiments were repeated at least three times. The motility complementation further verified an intact fusion protein (Fig. S6).

**Growth and confocal microscopy of protein fusions.** Cultures harboring the RelA-YFP construct were grown in BM2 (at 250 rpm) to an $OD_{600}$ of 0.3 prior to induction with 5% L-arabinose. After 30 min of induction, cultures were split and one received with 1 mM SHX for another 30 min. Stationary-phase cultures at an OD of 2.5 ± 0.3 were induced with 5% L-arabinose for 30 min. Cultures harboring SpoT-GFP were grown to an $OD_{600}$ of 0.3 before induction with 5% L-arabinose for 30 min. Cultures were washed

one time with phosphate-buffered saline (PBS) (8,000 rpm, 5 min) before resuspending in BM2 with or without a carbon source for 30 min. Iron starvation was induced with 1 mM dipyridyl for 30 min.

To visualize the fluorescent protein localization in cells, the fluorescent proteins were cross-linked to achieve minimal cell movement and preserve the physiological state. Cells were cross-linked with 3.7% formaldehyde at room temperature for 1 h. Cells were washed twice with PBS by centrifugation (8,000 rpm, 5 min) and the subsequent pellet resuspended in 1 ml of PBS. Cells were visualized on a Zeiss microscope.

**Growth, induction, and RNA isolation for qRT-PCR and RNA-Seq.** *P. aeruginosa* PAO1 wild-type, $\Delta relA$ $\Delta spoT$ double mutant, as well as *relA*- and *spoT*-overexpressing strains were grown to an $OD_{600}$ of 0.5 in 2YT broth. Induction experiments were carried out with either 1 mM SHX to chemically induce stringent conditions in the wild type or via overexpression of *relA* or *spoT* from pHERD20T through the addition of 1% L-arabinose to the culture medium. Induced cultures were left for 30 min at 37°C with shaking (250 rpm).

Bacteria were harvested in RNAprotect bacterial reagent (Qiagen) by centrifugation (13,000 rpm, 2 min). Total RNA was isolated using the RNeasy minikit (Qiagen) by following the manufacturer's instructions. The obtained RNA was DNase treated (Ambion/Life Technologies) and subsequently quantified using a NanoDrop ND-2000 spectrophotometer (Thermo Fischer Scientific), and RNA integrity determined by agarose gel electrophoresis.

For qRT-PCR experiments, high-quality RNA was reverse transcribed and amplified with a Roche LightCycler 96 instrument, in combination with the qScript one-step SYBR green qRT-PCR kit (QuantaBio), according to the manufacturer's protocol. Template RNA (5 ng/sample) was used in a standard 25-$\mu$l qRT-PCR with specific primers. Each sample was analyzed for gene expression in at least triplicate. Quantification of mRNA transcripts was performed by the comparative threshold cycle method (85) using the *16S* gene as a normalizer.

For RNA-Seq experiments, rRNA was further depleted using the RiboZero bacterial kit (Illumina). Library preparation was done with the KAPA stranded total RNA kit (Kapa Biosystems), and sequencing was performed on the Illumina HiSeq2500 instrument at the University of British Columbia's Sequencing and Bioinformatics Consortium (generating single-end reads of 1× 100 bp). The read quality and alignment of sequencing samples were carried out as previously described (84). Briefly, FastQC v0.11.6 and MultiQC v1.6 were used for quality, STAR v2.6 for the alignment of transcriptomic reads to the PAO1 reference genome (obtained from the *Pseudomonas* Genome Database [58]), and read counts generated using HTSeq v0.11.2. Library sizes had a minimum of 1.2 million, median of 3.1 million, and maximum of 7.6 million uniquely mapped reads. Differentially expressed (DE) genes between the double mutant and wild type were determined using DESeq2 v1.24.0, with thresholds of adjusted *P* values of ≤0.05 and absolute fold change of ≥1.5.

**Visualization of RNA-Seq data.** Relative expression of DE genes was plotted using the circular visualization software package CIRCOS (86). Since we found stretches of genes that were dysregulated in the same direction, we used an arbitrary cutoff of five consecutive genes to extract gene information for these regions. The rationale for using at least five genes in a row was that four genes almost revealed 100 clusters and six in a row also showed 34 clusters.

**Regulatory elements and regulator enrichment.** Operon information for *P. aeruginosa* PAO1 was downloaded from DOOR2 (87) and gene annotations obtained from the *Pseudomonas* Genome Database (58). The following methodology was then applied to all 1,673 DE genes identified in the RNA-Seq experiment. The transcriptional start site (TSS) of each gene was obtained from Gill et al. (66). For any TSS type denoted as antisense, the strand for that gene was switched. In case there was no TSS available, the start codon as listed in the genome annotations was used. The list of 1,673 genes then was filtered to include the first gene from a given operon (or single genes for those not in an operon), yielding 1,201 genes. Once the starting location had been determined, the R package BSGenome v1.48.0 (88) was used to extract the 250-bp upstream region for each of the 1,201 genes. These sequences were then submitted to MEME (89) for identification of novel motifs with a significance threshold E value of ≤0.05. The program was set to find up to five motifs, while all other settings were left at their defaults. These five *de novo* motifs found by MEME were then submitted to TomTom (90) to identify potential matches to characterized motifs and their corresponding regulators within the CollecTF database (67). Matches identified by TomTom were considered significant with a *q* value of ≤0.5.

The list of all PAO1 regulators and their controlled genes (regulon) was downloaded from CollecTF (67) or recent Chip-Seq manuscripts as indicated in the text. This list of regulons was then tested for enrichment in the list of 1,673 DE genes using Fisher's exact test, implemented via a custom script in R. Multiple test correction was performed using the Benjamin-Hochberg (BH) method. Significance of enrichment for each regulon was determined using a threshold of BH-corrected *P* values of ≤0.05.

**Functional enrichment of DE genes.** Enrichment of GO terms was performed using GofuncR (Steffi Grote, 2018), testing the DE genes against a custom set of GO annotations downloaded from the *Pseudomonas* Genome Database (58). The full list of 1,673 DE genes was split into up- and downregulated genes, with GO enrichment being performed independently on each of these sets. Results were filtered using a significance threshold of family-wise error rate (FWER) of ≤0.1. Enrichment of KEGG pathways was done using Gage v2.3.0 (91) on the full list of 1,673 DE genes. Results were filtered for significance based on q value of ≤0.2.

Enrichment of cellular functions, based on manually curated lists from various sources (Data Set S2), was performed on the full list of 1,673 DE genes using Fisher's exact test, implemented via a custom script in R. Multiple test correction was performed using the BH method and filtered on a significance of ≤0.05.

**Growth curve experiments, pyoverdine production, and pyocyanin measurements.** PAO1 strains were grown overnight (16 to 18 h) in 2YT medium at 37°C with shaking (250 rpm). The overnight culture was resuspended in 2YT broth to an $OD_{600}$ of 0.1, and then 200 $\mu$l was transferred to a flat-bottom 96-well polystyrene microtiter plate (Corning) and incubated at 37°C with continuous fast linear shaking at 567 cpm in a microplate reader (Synergy H1; BioTek). $OD_{600}$ and fluorescence readings were taken every hour over a 24-h period, as previously described (84). Experiments were performed three times with at least three technical replicates.

Pyocyanin production of PAO1 and PAO1 Δ*relA* Δ*spoT* strains was measured as previously described (84). Briefly, cells were grown in LB overnight and subsequently washed in SCFM broth, adjusted to an $OD_{600}$ of 0.1, and further cultivated at 37°C with aeration (250 rpm) for 20 h, and pyocyanin was extracted from filter-sterilized supernatants. Experiments were performed three times with at least two technical replicates.

**Swarming motility assays.** Swarming motility was examined on KB plates containing 0.4% agar. Strains were adjusted in KB medium to an $OD_{600}$ of 0.1 and incubated for 24 h at 37°C as previously described (82). All experiments were performed at least three times.

**Adherence experiments.** Adherence experiments were performed as previously described (84). Briefly, strains were streaked onto 2YT agar plates and grown overnight at 37°C. Bacteria were scraped from the plates and resuspended in 2YT medium to an $OD_{600}$ of 0.5, and 100 $\mu$l was added into polystyrene microtiter plates (Falcon) and incubated at room temperature for 1 h. Each well then was washed and adhered cells stained with crystal violet. Afterwards, plates were washed and crystal violet dissolved in 70% ethanol at room temperature, and absorbance was measured at 595 nm with a microplate reader (Synergy H1; BioTek). Data analysis was performed to calculate the mean and standard deviation after removal of outliers that were more than one standard deviation from the mean. Data were further normalized to the wild type. Experiments were performed at least three times with up to six technical replicates.

**Ethics statement.** Animal experiments were performed in accordance with The Canadian Council on Animal Care (CCAC) guidelines and were approved by the University of British Columbia Animal Care Committee (certificate number A14-0363).

**Cutaneous mouse infection model.** Mice used in this study were female outbred CD-1. All animals were purchased from Charles River Laboratories (Wilmington, MA), were 7 weeks of age, and weighed about 25 ± 3 g at the time of the experiments. One to 3% isoflurane was used to anesthetize the mice. Mice were euthanized with carbon dioxide. The cutaneous mouse abscess infection model was performed as described earlier (92). Briefly, *P. aeruginosa* PAO1 and its alkaline protease AprA-deficient mutant were grown to an $OD_{600}$ of 1.0 in 2YT broth, subsequently washed twice with sterile PBS, and further adjusted to $5 \times 10^8$ CFU/ml. A 50-$\mu$l bacterial suspension was injected into the right side of the dorsum. The progression of the infection was monitored daily and mice euthanized that reached the humane endpoint. Abscess lesion size (visible dermonecrosis) was measured on day three. Skin abscess tissues were excised (including all accumulated pus) and homogenized in 1 ml sterile PBS using a Mini-Beadbeater-96 (Biospec products) for 5 min, and bacterial counts were determined by serial dilution. Experiments were performed at least 2 times independently with 3 to 5 animals per group.

**Data availability.** All fastq and count files are available under Gene Expression Omnibus (GEO) accession number GSE147132. The full list of differentially expressed genes is included in Data Set S1.

## SUPPLEMENTAL MATERIAL

Supplemental material is available online only.

**FIG S1**, EPS file, 0.4 MB.
**FIG S2**, EPS file, 0.4 MB.
**FIG S3**, EPS file, 0.1 MB.
**FIG S4**, EPS file, 0.2 MB.
**FIG S5**, EPS file, 1.5 MB.
**FIG S6**, EPS file, 0.4 MB.
**FIG S7**, EPS file, 0.8 MB.
**TABLE S1**, DOCX file, 0.02 MB.
**DATA SET S1**, XLSX file, 0.6 MB.
**DATA SET S2**, XLSX file, 0.2 MB.

## ACKNOWLEDGMENTS

D.P. is supported by an Alexander von Humboldt–Feodor Lynen Postdoctoral Fellowship, a Cystic Fibrosis Canada Postdoctoral Fellowship, and a Research Trainee Award from the Michael Smith Foundation for Health Research. R.E.W.H. is supported by Canadian Institutes from Health Research grant FDN-154287 and holds a Canada Research Chair and UBC Killam Professorship.

We thank Caleb Ritchie for his help with adherence- and motility-based experiments and Leo Liu for his help with measuring pyoverdine and pyocyanin production.

We have no competing interests to declare.

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
