## [Reviewer comments · mSystems]

The stringent stress response controls proteases and global regulators under optimal growth conditions in *Pseudomonas aeruginosa*

Daniel Pletzer, Travis Blimkie, Heidi Wolfmeier, Yicong Li, Arjun Baghela, Amy Lee, Reza Falsafi, and Robert Hancock

Corresponding Author(s): Daniel Pletzer, University of Otago

Review Timeline:

Submission Date:	June 2, 2020
Editorial Decision:	July 7, 2020
Revision Received:	July 13, 2020
Accepted:	July 14, 2020

Editor: David Cleary

Reviewer(s): The reviewers have opted to remain anonymous.

Transaction Report:

DOI: <https://doi.org/10.1128/mSystems.00495-20>

July 7, 2020

Dr. Daniel Pletzer
University of Otago
Microbiology and Immunology
720 Cumberland St
PO Box 56
Dunedin 9054
New Zealand

Re: mSystems00495-20 (The stringent stress response controls proteases and global regulators under optimal growth conditions in *Pseudomonas aeruginosa*)

Dear Dr. Daniel Pletzer:

I have now received reviews on your submitted manuscript. Overall both reviewers were positive and have only raised some minor points for you to address.

Below you will find the comments of the reviewers.

To submit your modified manuscript, log onto the eJP submission site at <https://msystems.msubmit.net/cgi-bin/main.plex>. If you cannot remember your password, click the "Can't remember your password?" link and follow the instructions on the screen. Go to Author Tasks and click the appropriate manuscript title to begin the resubmission process. The information that you entered when you first submitted the paper will be displayed. Please update the information as necessary. Provide (1) point-by-point responses to the issues raised by the reviewers as file type "Response to Reviewers," not in your cover letter, and (2) a PDF file that indicates the changes from the original submission (by highlighting or underlining the changes) as file type "Marked Up Manuscript - For Review Only."

Due to the SARS-CoV-2 pandemic, our typical 60 day deadline for revisions will not be applied. I hope that you will be able to submit a revised manuscript soon, but want to reassure you that the journal will be flexible in terms of timing, particularly if experimental revisions are needed. When you are ready to resubmit, please know that our staff and Editors are working remotely and handling submissions without delay. If you do not wish to modify the manuscript and prefer to submit it to another journal, please notify me of your decision immediately so that the manuscript may be formally withdrawn from consideration by mSystems.

To avoid unnecessary delay in publication should your modified manuscript be accepted, it is important that all elements you upload meet the technical requirements for production. I strongly recommend that you check your digital images using the Rapid Inspector tool at <http://rapidinspector.cadmus.com/RapidInspector/zmw/>.

Sincerely,

David Cleary

Editor, mSystems

Journals Department
Reviewer comments:

Reviewer #1 (Comments for the Author):

Overall this was a well written and argued paper. There were a couple of points that should be clarified.

Line 173 "Since both RelA and SpoT were obviously present during logarithmic growth in nutrient-rich conditions (SI Appendix, Fig S1) (17, 18)" Not sure that this conclusion is based on Figure S1 which does not mention spoT.

For the section on the aprA mutant as a downstream effector of the stringent response and required for full virulence, I thought there was some need for clarity. First whether this section really fit in this manuscript or if it should be in future work on the downstream mechanisms of the stringent response as suggested in lines 431-434.

Line 32-34: "Investigation of an aprA mutant in a murine skin infection model, showed increased survival rates of the aprA mutant consistent with previous observations that stringent-response mutants have reduced virulence." I think should be "...increased survival rates of the mice infected with the aprA mutant..." Likewise I suggest lines 303-305 "However, after three days there was a significant enhancement of survival (77%) of the aprA mutant cf. the wild-type PAO1 (~33% survival of mice) (Figure 3D)." This might be clearer as "... a significant enhancement of survival (77%) of the mice infected with the aprA mutant over the mice infected with the wild type PAO1 (~33% survival...". Finally on lines 983-84 the authors used the Gehan-Breslow-Wilcoxon test which is appropriate but the more standard test is the log rank test and might be more appropriate for this data. At least I would encourage the authors to make a statement as to whether their p-value of 0.03 is significant for this study.

Reviewer #2 (Comments for the Author):

In this study, the authors carry out a comprehensive analysis of the transcriptional profile of a stringent response mutant under non-stressful growth conditions, with special emphasis in virulence factors and other global regulators. The authors describe in detail the main clusters of genes that were differentially expressed under these conditions and how that could contribute to adaptation during infection. Overall, the work is very relevant to the study of *P. aeruginosa* transcriptional regulation by ppGpp, which is a gap that has existed for years, in comparison to what is already known for *E. coli* and other bacteria.

Some questions and comments that the authors may wish to consider:

1. Appendix 1. "Together, these results suggested that both the *relA* and *cysM* promoter are involved in activation of the *RelA* synthase gene expression during amino acid starvation in *P. aeruginosa*". - The evidence shows that both promoters are active but not that both induce the expression of *relA* (in particular for the *cysM* promoter). To further support this statement, the authors could include the mapping of the reads from the transcriptomics data to show that *relA* is being transcribed by both promoters.
2. Figure 1. Why did the signal from *RelA*/*SpoT* decrease in SR-inducing conditions? And why is the signal from *SpoT* so different during stationary phase in comparison to the other conditions, when in theory the stringent response would be activated during this phase?
3. Line 191: "Gene Ontology (GO) and pathway enrichment (KEGG) were used for functional enrichment analysis". - The authors do not mention what strain was used as baseline in the RNA seq analysis (PAO1 WT or the SR mutant). Further in the text (line 197) this is clarified, but at the beginning of the text is hard to interpret if the pathways are up/down regulated in the mutant or in the wild-type. Please clarify this at the beginning of the paragraph.
4. Line 219: "We also observed that differential expression often involved co-expressed neighbouring genes". - Any thoughts on why this might be happening?
5. Line 253: "We observed down regulation in the $\Delta relA \Delta spoT$ stringent response double mutant of *rhII* (-3.0 fold), *rhIR* (-2.6 fold), *lasR* (-1.6 fold), *rsaL* (-3.0 fold), *pqsH* (-2.0 fold) and *pqsL* (-3.6 fold; cf. *pqsA-E* that were 3.9-4.6 fold upregulated)". - How was the expression of *pqsR*? Since it has been shown that the PQS system is up-regulated in the SR mutant.
6. SI Appendix, Fig S4. Please indicate the statistical test that was used.
7. Dataset 1. Minor point: I am not sure if it is because of the use of certain number of decimals, instead of using scientific notation, but an adjusted p-value of 0 seems odd.
8. Table 1. "Differential effect of loss ($\Delta relA \Delta spoT$) and overexpression of ppGpp (+SHX/*relA*++)" I suggest that the authors replace overexpression with "overproduction".
9. According to the dataset 4, *lasI* is not significantly differentially expressed in the mutant. Why do you think *lasI* expression is not reduced, given that *lasR* is downregulated?

10. Figure 3. LasB was previously mentioned in the paper to be downregulated in the SR mutant (line 242), which can be also found in the dataset 1, however it does not appear in the figure 3 along with the other proteases, is there any reason for that?
11. Table 1. For the qRT-PCR experiments, what control was used? (i.e. a gene that is not regulated by stringent response and therefore its expression should not change in the absence of ppGpp). According to the methods, the authors used the gene 16S to normalise the qRT-PCR data, but one would have thought that ppGpp might affect the expression of the 16S rRNA?
12. Table 1. How would you explain that algR was downregulated both in SR mutant and WT SHX-induced by the same amount, but upregulated after overexpression of RelA?
13. Line 315: "The zinc-dependent protease PA0277 (5.2 fold downregulated by SHX induction, 9.9-fold upregulated in the mutant) is directly controlled by the post-transcriptional regulator RsmA (59), and the expression of rsmA is directly activated by AlgR (60) and indirectly via GacA through RsmY and RsmZ (61). All of these were further controlled by the stringent stress response under normal conditions." - I could not find RsmA/Y/Z in the dataset 1 for the differentially regulated genes. Please amend.
14. I might have missed it, but it would be good to see ppGpp levels measured, at the very least in the WT and in the null mutant (and SHX-treated WT?). This is easy to do.
15. Finally, a nod to one of the previous studies carried out on a virulence factor-secreting QS Gram-negative organism, *Erwinia atrosepticum*, should be made, with some discussion of how that earlier work maps onto the current findings ["Virulence in *Pectobacterium atrosepticum* is regulated by a coincidence circuit involving quorum sensing and the stress alarmone, (p)ppGpp." By Bowden et al, *Mol Micro* (2013)].

Dear Editor,

We wanted to thank you and all reviewers for their feedback. We have further responded in plain type for each issue in turn with the reviewer's comments in italics. Modifications into the manuscript text are red in the marked copy of the manuscript.

Sincerely Yours,

Pletzer Daniel, PhD

and

R.E.W. Hancock,

UBC Killam Professor and Canada Research Chair

Responses to Questions

Reviewer #1 (Comments for the Author):

Overall this was a well written and argued paper. There were a couple of points that should be clarified.

Line 173 "Since both RelA and SpoT were obviously present during logarithmic growth in nutrient-rich conditions (SI Appendix, Fig S1) (17, 18)" Not sure that this conclusion is based on Figure S1 which does not mention spoT.

Thanks for spotting that. We have added the *rpoZ-spoT*.GFP reporter fusion data as new Supplementary Figure S2.

For the section on the aprA mutant as a downstream effector of the stringent response and required for full virulence, I thought there was some need for clarity. First whether this section really fit in this manuscript or if it should be in future work on the downstream mechanisms of the stringent response as suggested in lines 431-434.

The data on the metalloprotease AprA complements our novel findings that proteases are downstream effectors of the stringent stress response and that the analysis under non-stressful conditions has value in translation to *in vivo* infection. While we agree that the *aprA* data is a bit of stretch from the main RNA-Seq data analysis, we think that it has merit to include to show the relevance of our study from a translational aspect.

Line 32-34: "Investigation of an aprA mutant in a murine skin infection model, showed increased survival rates of the aprA mutant consistent with previous observations that stringent-response mutants have reduced virulence." I think should be "...increased survival rates of the mice infected with the aprA mutant..."

We have added 'mice infected with' accordingly.

Likewise I suggest lines 303-305 "However, after three days there was a significant enhancement of survival (77%) of the aprA mutant cf. the wild-type PAO1 (~33% survival of mice) (Figure 3D)." This might be clearer as "... a significant enhancement of survival (77%) of the mice infected with the aprA mutant over the mice infected with the wild type PAO1 (~33% survival....".

The change was made accordingly.

Finally on lines 983-84 the authors used the Gehan-Breslow-Wilcoxon test which is appropriate but the more standard test is the log rank test and might be more appropriate for this data. At least I would encourage the authors to make a statement as to whether their p-value of 0.03 is significant for this study.

We have added the Log-rank (Mantel-Cox) test and indicated significance $p\text{-value} \leq 0.05$ accordingly. Both tests showed significance at 0.032 (GHB) and 0.025 (MC).

Reviewer #2 (Comments for the Author):

In this study, the authors carry out a comprehensive analysis of the transcriptional profile of a stringent response mutant under non-stressful growth conditions, with special emphasis in virulence factors and other global regulators. The authors describe in detail the main clusters of genes that were differentially expressed under these conditions and how that could contribute to adaptation during infection. Overall, the work is very relevant to the study of *P. aeruginosa* transcriptional regulation by ppGpp, which is a gap that has existed for years, in comparison to what is already known for *E. coli* and other bacteria. Some questions and comments that the authors may wish to consider:

1. Appendix 1. "Together, these results suggested that both the *relA* and *cysM* promoter are involved in activation of the *RelA* synthase gene expression during amino acid starvation in *P. aeruginosa*". - The evidence shows that both promoters are active but not that both induce the expression of *relA* (in particular for the *cysM* promoter). To further support this statement, the authors could include the mapping of the reads from the transcriptomics data to show that *relA* is being transcribed by both promoters.

We have not performed any mapping to non-coding regions in this study. The study by Wurtzel et al (1) already showed that *relA* is driven by two independent promoters with a continuous transcript (see Figure below). This work has been referenced in Appendix 1 in the first sentence.

(1) Wurtzel O, Yoder-Himes DR, Han K, et al. The single-nucleotide resolution transcriptome of *Pseudomonas aeruginosa* grown in body temperature. *PLoS Pathog.* 2012;8(9):e1002945. doi:10.1371/journal.ppat.1002945

2. Figure 1. Why did the signal from *RelA*/*SpoT* decrease in SR-inducing conditions? And why is the signal from *SpoT* so different during stationary phase in comparison to the other conditions, when in theory the stringent response would be activated during this phase?

Proteins that localize in one foci show stronger fluorescence, hence higher intensity as compared to a signal that is equally distributed inside a cell. Since the foci became more homogenous throughout the cell upon stringent induction, the signal became more evenly distributed and therefore the intensity dropped. *SpoT* is active throughout the cell cycle and stationary phase (we have added the *rpoZ*-*spoT*-GFP fusion data as new Supplementary Figure S2). The transition to early stationary phase will slowly increase ppGpp levels, which appears to not trigger *SpoT* activity. *SpoT* might therefore rather be activated in late stationary phase to degrade high levels of ppGpp; however, this was not tested in this study.

3. Line 191: "Gene Ontology (GO) and pathway enrichment (KEGG) were used for functional enrichment analysis". - The authors do not mention what strain was used as baseline in the RNA seq analysis (PAO1 WT or the SR mutant). Further in the text (line 197) this is clarified, but at the beginning of the text is hard to interpret if the pathways are up/down regulated in the mutant or in the wild-type. Please clarify this at the beginning of the paragraph.

Thanks. We modified the sentence accordingly:

"Gene Ontology (GO) and pathway enrichment (KEGG) were used for functional enrichment analysis of differentially expressed genes identified when comparing the PAO1 stringent response mutant to the PAO1 wild-type."

4. Line 219: "We also observed that differential expression often involved co-expressed neighbouring genes". - Any thoughts on why this might be happening?

We are not entirely sure why this is happening, but it could indicate that co-regulated operons (regulons) might have additional, hitherto-unknown genes expressed in adjacent locations on the chromosome. It would make sense that such genes might be clustered since transcription preferentially occurs in regions of the genome that are unwound during replication; thus genes localized to particular genomic neighbourhoods would be easier to co-express. This however is quite speculative and any confirmation would require more extensive studies involving global regulators and additional *Pseudomonas* RNA-Seq data to identify the complement and genomic location of particular regulons.

5. Line 253: "We observed down regulation in the $\Delta relA\Delta spoT$ stringent response double mutant of *rhII* (-3.0 fold), *rhIR* (-2.6 fold), *lasR* (-1.6 fold), *rsaL* (-3.0 fold), *pqsH* (-2.0 fold) and *pqsL* (-3.6 fold; cf. *pqsA-E* that were 3.9-4.6 fold upregulated)". - How was the expression of *pqsR*? Since it has been shown that the PQS system is up-regulated in the SR mutant.

PqsR (PA1003) was -1.25 downregulated; however, the adjusted p-value (0.27) indicated low confidence in this data.

6. SI Appendix, Fig S4. Please indicate the statistical test that was used.

We have added the test accordingly.

7. Dataset 1. Minor point: I am not sure if it is because of the use of certain number of decimals, instead of using scientific notation, but an adjusted p-value of 0 seems odd.

We apologize for that; these values were so low that they were effectively zero. The updated Dataset 1 contains the adjusted p-values in scientific format now.

8. Table 1. "Differential effect of loss ($\Delta relA\Delta spoT$) and overexpression of *ppGpp* (+SHX/*relA*++)" I suggest that the authors replace overexpression with "overproduction".

The change was made accordingly.

9. According to the dataset 4, *lasI* is not significantly differentially expressed in the mutant. Why do you think *lasI* expression is not reduced, given that *lasR* is downregulated?

We have not performed follow up experiments on *lasI*, but given the RNA-Seq data, *lasI* (PA1432) was -1.32 down-regulated, which is very close to our cut-off threshold of ± 1.5 . The downregulation of *lasR* (PA1430) was -1.57 fold; so both were actually downregulated in our study.

10. Figure 3. *LasB* was previously mentioned in the paper to be downregulated in the SR mutant (line 242), which can be also found in the dataset 1, however it does not appear in the figure 3 along with the other proteases, is there any reason for that?

Thanks for spotting this in Fig 3. The reason why *lasB* was missed was because of our search terms at pseudomonas.com. We used 'protease' and 'peptidase' which did not reveal *lasB* as a protease. MEROPS also does not list *lasB* as a peptidase for PAO1. We have now added *lasB*.

11. Table 1. For the qRT-PCR experiments, what control was used? (i.e. a gene that is not regulated by stringent response and therefore its expression should not change in the absence of *ppGpp*). According to the methods, the authors used the gene 16S to normalise the qRT-PCR data, but one would have thought that *ppGpp* might affect the expression of the 16S rRNA?

We always check multiple genes for their suitability to use in normalization under all test conditions. In this case, we used the 16S gene, which demonstrated stable expression with an average standard deviation of 19% among all Ct values from all samples (obtained from 10 different qRT-PCR runs and ~30 samples). The overexpression of *relA* or overproduction of *ppGpp* (SHX) did not affect the expression of the 16S gene.

12. Table 1. How would you explain that *algR* was downregulated both in SR mutant and WT SHX-induced by the same amount, but upregulated after overexpression of *RelA*?

Based on our data (Table 1) we would argue that the expression of *algR* was only weakly downregulated in the stringent response mutant (-1.4 fold) and upon SHX induction (-1.2 fold), i.e. below our confidence level cut-off. We hypothesize that the overexpression of *relA* most likely mimics ppGpp effects during normal growth, and induction via SHX would reflect an alarm response to amino acid starvation.

13. Line 315: "The zinc-dependent protease PA0277 (5.2 fold downregulated by SHX induction, 9.9-fold upregulated in the mutant) is directly controlled by the post-transcriptional regulator RsmA (59), and the expression of *rsmA* is directly activated by AlgR (60) and indirectly via GacA through RsmY and RsmZ (61). All of these were further controlled by the stringent stress response under normal conditions." - I could not find *RsmA/Y/Z* in the dataset 1 for the differentially regulated genes. Please amend.

Thanks for that. Indeed, we have not looked at the expression of non-coding RNAs in this study. Therefore, we have changed the second sentence to 'The response regulators AlgR and GacA were controlled by the stringent stress response under normal conditions'.

14. I might have missed it, but it would be good to see ppGpp levels measured, at the very least in the WT and in the null mutant (and SHX-treated WT?). This is easy to do.

We have not measured ppGpp levels in this study. However, we have previously determined ppGpp levels in the *P. aeruginosa* PAO1 wildtype, stringent response mutant, and upon induction with SHX (1).

- (1) de la Fuente-Núñez C, Reffuveille F, Haney EF, Straus SK, Hancock RE. Broad-spectrum anti-biofilm peptide that targets a cellular stress response. *PLoS Pathog.* 2014;10(5):e1004152. Published 2014 May 22. doi:10.1371/journal.ppat.1004152

15. Finally, a nod to one of the previous studies carried out on a virulence factor-secreting QS Gram-negative organism, *Erwinia atrosepticum*, should be made, with some discussion of how that earlier work maps onto the current findings ["Virulence in *Pectobacterium atrosepticum* is regulated by a coincidence circuit involving quorum sensing and the stress alarmone, (p)ppGpp." By Bowden et al, *Mol Micro* (2013)].

Thank you. The suggested study nicely supports our data showing that proteases are downstream effectors of the stringent response. We have included this publication and mention it as follows: "Intriguingly, Bowden et al (62) described a link between the stringent stress response and the loss of *rsmA* expression, which restored protease production in the plant pathogen *Erwinia atrosepticum*. This further supports our conclusions that proteases are downstream effectors of the stringent response in *P. aeruginosa*."

July 14, 2020

Dr. Daniel Pletzer
University of Otago
Microbiology and Immunology
720 Cumberland St
PO Box 56
Dunedin 9054
New Zealand

Re: mSystems00495-20R1 (The stringent stress response controls proteases and global regulators under optimal growth conditions in *Pseudomonas aeruginosa*)

Dear Dr. Daniel Pletzer:

Thank you for submitting your responses to the reviewer comments and the revised manuscript. I am pleased to say that your manuscript has been accepted, and I am forwarding it to the ASM Journals Department for publication. For your reference, ASM Journals' address is given below. Before it can be scheduled for publication, your manuscript will be checked by the mSystems senior production editor, Ellie Ghatineh, to make sure that all elements meet the technical requirements for publication. She will contact you if anything needs to be revised before copyediting and production can begin. Otherwise, you will be notified when your proofs are ready to be viewed.

Sincerely,

David Cleary
Editor, mSystems

Journals Department
Dataset 1: Accept
Fig. S1: Accept
Fig. S6: Accept
Fig. S5: Accept
Fig. S4: Accept
Table S1: Accept
Dataset 2: Accept
Fig. S2: Accept
Fig. S3: Accept
Fig. S7: Accept